# What Is a School Farm? Results of a Scoping Review

**DOI:** 10.3390/ijerph20075332

**Published:** 2023-03-30

**Authors:** Sammy A. Blair, Gabrielle Edwards, Katharine Yu, Eduardo Jovel, Lisa Jordan Powell, Kerry Renwick, Annalijn I. Conklin

**Affiliations:** 1Faculty of Land and Food Systems, University of British Columbia, Vancouver, BC V6T 1Z4, Canada; 2Faculty of Education, University of British Columbia, Vancouver, BC V6T 1Z4, Canada; 3Center for Human and Environmental Sustainability, Sweet Briar College, Sweet Briar, VA 24595, USA; 4Collaboration for Outcomes Research and Evaluation, Faculty of Pharmaceutical Sciences, University of British Columbia, Vancouver, BC V6T 1Z3, Canada; 5Centre for Health Evaluation and Outcome Sciences, Providence Healthcare Research Institute, St. Paul’s Hospital, Vancouver, BC V6Z 1Y6, Canada

**Keywords:** school farm, food education, food access, school food, agricultural education, food literacy, food security, food systems, vocational agriculture

## Abstract

As school farms become more prominent programs to teach food education, research is needed to support school farms’ implementation and sustainability. This scoping review included 94 articles on school farms from three bibliometric databases covering broad international literature. Vocational agricultural training, animal husbandry, and crop production were common characteristics of school farms across 103 years of publications. Themes of sustainability, healthy eating, and food systems were more prominent in recent literature. Peer-reviewed studies (1985–2019) provided some empirical research showing school farms’ impact on students. This review discusses school farms’ structures and objectives as promising food education and production programming.

## 1. Introduction

Research shows young adults who leave secondary school without consistent food education lack knowledge of basic nutrition, food skills, food systems, everyday food practices, and food production [1,2]. Links have been established between food system knowledge and protecting the environment [1,3,4]. There is also recent research on the importance of socio-cultural competency as a part of food literacy to support food practices, healthcare, and education involving food systems [5] and diet-related diseases [6,7]. Healthy School BC defines food literacy as food-related knowledge, attitudes and skills which include understanding the linkages between food, health and wellbeing, knowing how to select nutritious foods and comprehending what constitutes a healthy diet [8]. Providing students with a capacity for food literacy, or the understanding, skills, and behaviours to participate in an interconnected food system, will promote understanding and knowledge that is critical for healthier and more just food systems [9], community food security, and food sovereignty of current and future generations [9,10,11,12].

When individuals develop an understanding about what they eat and how their food is produced, there is potential for different levels of engagement and action when making food choices and interacting with the food system [4,9,11]. Some of these choices reflect the social, cultural, economic, and political factors affecting people’s access, control, and thus impact on the food system [5,13,14]. The expected benefits of food literacy education have inspired schools to prioritize nutrition education and also expand it to include broader food system topics of agriculture, ecology, social, and economic food- and health-related topics as essential knowledge for today’s complex food environments [1,5,15].

There are many different delivery methods of food literacy education. Examples include: educators’ use of school gardens; classroom cooking programs; class connections with local farms; and units within home economics courses [11,16,17,18,19]. One example of a specific food education program is a school farm. School farms teach about the food system and engage students through direct experiences of agricultural production on farms managed by or in collaboration with schools. School farms use experiential learning pedagogies to contribute to food literacy and support food education to ultimately build community food security [10]. According to recent literature in the 21st century, school farms are also known to engage students who may be marginalized in their communities [20,21,22] because the programs offer a venue for students from different backgrounds and cultures to find purpose, connection, and skills in an educational environment outside the traditional classroom setting [12].

School farms may have the potential to involve diverse students, create opportunities for new connections and skills, provide food for school meal programs [23,24,25,26], and engage students in academic subjects using food system frameworks and related pedagogies. However, foundational knowledge on common objectives and activities of school farms as unique farm-to-school programming is needed to better understand their potential [27]. Specifically, in Canada, as organizations and schools engage in a burgeoning school farm movement, community feedback emphasised a need for more research on how to define school farms to create consistency and replicability in their development [28]. Thus, the purpose of this scoping review was to clarify definitions in the literature by identifying key characteristics related to the concept of this food education intervention and to identify and analyse knowledge gaps to support future research in this area. Our research question was, ‘what is a school farm?’ This review will identify the types of available evidence on school farms and summarise key characteristics that help to better define a school farm. This review will support improved standardization for future research and policy since this study will result in a more unified understanding of self-defining school farm programs as unique food education interventions.

## 2. Methods

We performed a systematic search of both scholarly and grey literature on school farms using three different bibliometric databases: CAB Abstracts & Global Health, Web of Science Core Collection, and Education Source. Our protocol followed the PRISMA extension method for scoping reviews (PRISMA-ScR) [29]. These three databases were chosen through pilot searches based on their interdisciplinary specializations and their relevant search results. We deliberately did not include terms such as “school gardens”, or “farm-to-school” programming, so as to identify self-defining school farms as a unique and specific food education intervention. As a scoping review, we were also interested in understanding what authors in the field identified or understood to be the school farm movement which is happening predominantly across grades K-12, since their identities, structures, and objectives differ greatly from programs in university or post-secondary settings.

We applied a single, broad search term of “school farm*” after consultation with a reference librarian with subject-area specialization in agriculture and agroecology. The choice of single search term was purposeful to ensure consistency and replicability, and to capture a wide literature on education, science, health, and agriculture while remaining specific to self-defining school farm programs. No limitations on publication date, publication type (e.g., editorial, commentary, magazine, original paper, etc.), age group, or location were placed on the search in each database, and separate searches were performed by three independent reviewers (SB, GE, KY) in October 2021. We excluded publications describing other educational farms such as university farms (111 that did not identify as a “school farm”). Longstanding farms at universities and colleges, such as those at land grant institutions, have been studied for decades [30], but are not the same programs and do not use the name “school farm” as school farms today or those found in our preliminary search. Articles were excluded if they were not published in English, were book reviews, or did not have a direct reference to a characteristic or defining aspect of a school farm. Search results were retrieved and managed using 2022 Covidence web-based software. Uncertainties of publication eligibility were resolved through consensus among all reviewers. This scoping review was not registered and the protocol is available upon request to the corresponding author.

Titles and abstracts were screened to include any literature describing the characteristics and principles of a school farm (e.g., the definition, structure, goals, objectives, curriculum, etc.). Duplicate records were removed before eligible full-texts were retrieved and read in full for inclusion of publications meeting the eligibility criteria. References of all included publications were hand-searched and also screened by all three reviewers. Three reviewers extracted data from included publications using a standardized form in Excel with *a priori* headings. These were: citation, database, study location, the population of school farm, definitions of school farm, scale of school farm, program purpose/objectives, the structure of program, curriculum, and key findings. Duplicate extraction sheets were compiled by SB and verified by all three reviewers. We analysed the data through narrative synthesis and graphical display using Covidence. We used a bubble plot to display the frequency of publications (y-axis) and the time-period of publication over time (x-axis) for every type or category of literature (z-axis). Since our scoping review aimed to explore the literature on school farms, this review was not designed to quality assess the included papers.

## 3. Results

After removing duplicates (n = 78), our searches identified 748 unique records for the title and abstract screening. We retrieved 136 full-texts for eligibility assessment, and included 94 publications on school farms in the scoping review. We excluded 42 full-texts because the subject was not school farms (no detail on structure, programming, or definition) or the publication examined disease or illness from school farm proximity without information about the farm itself. Our pilot searches revealed that including all “school” synonyms resulted in unmanageably high numbers of results (Table 1); the majority of records were not relevant to this review. A summary of our search strategy is provided in Figure 1 below.

### 3.1. Summary of Scientific and Grey Literature on School Farms

A majority of the included studies (n = 75%) were editorials, newspapers, or magazine articles, and only 14 were peer-reviewed original studies [20,23,27,31,32,33,34,35,36,37,38,39] (Figure 2). The literature on school farms covered more than 100 years over two centuries. Several notable differences were between articles published during or just after the twentieth century versus those in the twenty-first century (Figure 3). From 1916 to 2009, articles were mostly editorials which accounted for 79% of the total literature. Of the empirical peer-reviewed publications (15% of total literature), 79% were published between 2010 and 2019. Editorials were published in two main sources: *Agricultural Education Magazine* and *The Times Educational Supplement*. Overall, most of the literature on school farms concerned programs in North America and the UK, although publications came from 12 other countries [20,21,23,24,25,31,32,37,38,40,41,42,43,44,45,46,47,48].

Across the literature, no clear definition of a school farm was provided, with one exception [20]. Corbett et al. defined five types of school farms as follows: (1) Agricultural education centres (AEC) are school farm operations with multiple activities (e.g., livestock, horticulture, aquaculture, etc.); (2) Specialized small farm operations (SSF) are smaller farms limited to specialized activities; (3) School gardens are horticultural operations with limited but engaging experiences for students; (4) Agricultural display/experience facilities (ADEF) are usually short-term off-site farm programming operations; and, (5) School-based land holdings (SBLH) are not for educational purposes but may provide income for schools. However, these five school farm types were not mentioned again in any of the literature.

Within the 94 publications, there were reports of a total 65 different school farms. The size of school farms varied widely. About 69% of farms in the literature noted their size reporting a range of 0.5–750 acres (Appendix A). School gardens are usually smaller than two acres [49], and only one school farm [26] was smaller than two acres (approximately 8000 m^2^). Additionally, while most literature focused on rural school farms, eight publications explicitly described their programs as being in urban environments [23,24,25,27,33,34,35,50].

### 3.2. Study Populations

Almost half of the literature described school farms for secondary students (grades 9–12) [20,23,26,33,34,36,43,46,48,50,51,52,53,54,55,56,57,58,59,60,61,62,63,64,65,66,67,68,69,70,71,72,73,74,75,76,77,78,79,80], however all education levels were reported in the literature. Notably, many publications did not report the grade of the program’s target population [42,44,45,81,82,83,84,85,86,87,88,89,90,91,92,93,94,95]. Regarding subpopulations, more than 41% of publications mentioned that the school farm programs were educational opportunities for boys specifically. No article mentioned school farms exclusively serving girls; however, one 2016 report from France indicated more girls than boys participated in the school farm program [46]. Another study in Switzerland [38] showed that girls found most school farm topics more interesting than the boys in the same program (Appendix A).

### 3.3. Early Characteristics of School Farms

Editorial articles accounted for 90% of literature from 1916–2009. School farms in this period were described as mainly being vocational agricultural programs for boys or young men. Only 8% of these articles mentioned girls or women as part of their student population. School farms were supported by a combination of school districts, student agricultural organizations (such as the National FFA Organization (Future Farmers of America)) [32,39,50,55,60,61,63,65,66,67,69,71,72,90,96,97,98]), which is a career and technical student organization that promotes middle and high school classes to include agricultural education, and on-site enterprises. 

In the literature published before 2000, the focus of school farms was on both animal husbandry [54,55,64,67,68,73,83,85,89,90,93] and crop production [54,55,64,65,67,68,73,83,85,90,93,99] education, with a secondary focus on experiential education, construction, farm maintenance, and farm mechanical training [41,59,61,74,75,83,89,91,100,101]. Interdisciplinary courses [50,51,52,66,82,102,103,104] on the school farms were mainly science [40,55,56,60,63,66,70,72,85,86,89,90,96,103] and math [66,81]. Consistent topics across school farm curricula included farm management [40,51,53,58,59,62,67,72,83,86,91,99,105,106], farm accounting and business [40,54,59,66,107], and environmental conservation [39,42,53,57,71,73,91,97,100,108]. Historically, school farms were described as being a mechanism for building and contributing to the local community through resource-sharing and financial opportunities for students [3,11,13,22,27,31,37,38,39,44,58,59,68,71].

### 3.4. Recent Literature on School Farms

Similar to earlier publications, literature on school farms from 2000–2019 described programs that mainly addressed animal husbandry [80,109,110] and crop production [78,80,111,112]. Although the literature still discussed vocational agricultural training, we found that recent literature after 1999 focused more on interdisciplinary learning [35,43], experiential education [20,22,26,32,33,77,109,113,114] and an expansion of interdisciplinary subjects to include writing [77,79,113] and technological education [77].

Publications in the new millennium reflected new objectives and curricular focus on nutrition and healthy eating. Except for a South African case study in 1977 that described school farms as a solution for malnutrition [32], the idea of nutrition or healthy eating was first mentioned in a 2008 publication [113]. The new focus on nutrition and healthy food education grew in prominence among school farm objectives throughout the early 2000s [20,23,26,33,38,94,115].

Additionally, environmentally focused objectives, including terms such as “ecology” and “sustainability”, were referenced among studies published in the 1980s and 1990s, mainly about Zuni people’s (North American Pueblo people in western New Mexico) cultural values on their school farm [71,74,75]. The literature from the 2000s described various themes regarding sustainable food education, including environmental and ecological values [38,46,113], education on sustainable farming, and understanding the food system [35,47]. Some newer literature has also identified food citizenship and social and political empowerment of students as an objective of school farms [20,21,34].

Although the necessity and value of community involvement remained a common theme across all of the school farm literature for the past two centuries [20,21,26,27,32,41,44,54,55,57,58,64,65,76,90,103,109,113,114,116], there was a shift in the roles of different community partners from the past. It was notable that literature from 2000 to 2019 rarely mentioned school districts [80,93] and their essential role in financial and managerial support that, in the past, was considered necessary to run a successful school farm. School farms in the 2000s appeared to be functioning as individual entities separate from national agriculture-related student organizations, with just a few mentions of FFA [2,32] and the Young Farmers Club [8,9,33,34], which are youth organizations that support student learning of agricultural skills as integral community partners. Finally, it was notable that literature from the 2000s did not report the gender of student participants, except for 11 studies [20,22,26,33,34,35,36,46,77,109,114] among which seven exclusively mentioned boys [20,22,26,33,35,36,114].

### 3.5. Empirical Research on School Farms

Of all 94 included publications, there were only 14 original studies providing empirical literature over a span of 40 years (1980–2019). Current research used quantitative and qualitative methods, and included multiple countries (Table 2). Most empirical studies were qualitative designs, commonly a case study, and used interview methodology for data collection. In addition to agricultural, experiential, and environmental education as school farm objectives (Table 2), some programs aimed to produce food for school meals [20,23,32], school feeding programs [23], or local markets [20]. The central proposition of school farms across these studies was that school farms create a microcosm for a food economy [32] while improving community engagement [20], agricultural literacy [22] and access to healthy affordable food [23].

Many of the peer-reviewed studies also suggested behavioural effects of school farm education on students. These studies showed that school farms promote responsibility [33], self-sufficiency [32], and self-efficacy [33] amongst students through the process of both group work and individual projects on school farms. Engaging students in inquiry-based and experiential learning is thought to promote leadership [40] and give students the skills to become agents of change [33] and active citizens [34] in their communities. While studies rarely reported the social demographics of their populations, two studies concluded that school farms gave educational and experiential opportunities to diverse groups of students [39,40] who may not have otherwise had access to agricultural, environmental, or experiential learning. In one study, 91% of parents supported school farms after seeing survey results of the education benefits of farming activities in kindergartens and primary schools [37].

In addition, studies reported that school farms diversified business strategies on farms and catered to the growing interest in local food and food-based issues [31] such as malnutrition [32] and food deserts [36]. On a systemic level, school farms are thought to have a “powerful symbolic presence” in rural communities because they provide relevant and vital curricula and pedagogy, and serve to support rural and educational policy-making [20]. We found no statements about unique characteristics of urban school farms and only six papers on urban school farms [23,24,25,27,33,35]. In other words, this scoping review revealed insufficient information about the similarities or differences to determine whether themes of rural education, policy, or community were shared or generalizable to urban school farms.

## 4. Discussion

This scoping review systematically examined a large body of literature on self-defined “school farms”, and found there is no clear understanding of what defines a school farm; moreover, robust evidence of their impact is still lacking. We found only one publication providing a definition of a ‘school farm’, and most of the published literature on this topic comes from editorials and other grey literature such as news and magazine articles; there is but 14 publications of peer-reviewed empirical research. Over three quarters of the grey literature came from only two publication sources: a magazine based in England and a journal in the US; the journal’s goal was to exchange professional news and views, be a sounding board for new ideas, and review publications. Thus, this literature has a strong Anglo-American bias, and its editorial nature begs the question whether local politics, policies, culture, and educational frameworks also bias the literature. Nevertheless, this scoping review showed that school farms are being used in many different geographic regions, ecological environments, and cultures to address a broad range of environmental, economic, health, agricultural, socio-cultural and educational issues.

According to the literature, unique features of school farms are that they have dedicated staff and volunteers to manage and maintain the farms, and they use an integrated curriculum that features food- and farm-related educational activities [27,95]. Overall, there was limited research evaluating the management, structure, role, or effects of school farms and a general belief that they are community-based solutions with the potential to create both academic and food education opportunities for students. There is a lack of rigorous empirical research on school farm programs for secondary students to inform development of future programs or education and food policy. Specifically, there are knowledge gaps on similarities/differences between rural and urban school farms, since only a handful of publications concerned urban school farms [23,24,25,27,33,35].

The prevalence of school farm themes related to nutritional, environmental, and community health in the new millennium aligns with trends in food literacy education [5,12]. Currently, food literacy education focuses on two main themes: (1) individual knowledge, choice, and skill; and (2) collective action, knowledge, and participation [4,5,11,19,20,117,118,119]. The first addresses a student’s food skills and knowledge of food, and the second refers to how students function as informed members of critical food contexts. Past school farms (prior to the year 2000) focused on vocational agricultural training, emphasizing animal husbandry and crop production and trade skills such as construction, farm maintenance, and farm mechanical training. By cotradt, current programs expanded upon these topics to include broader interdisciplinary ideas such as social and environmental food systems. This expansion to interdisciplinary and systems-thinking mirrors the evolution of food literacy themes more broadly. A potential explanation for this shift in school farm focus might be because food insecurity [120], diet-related chronic illness [121] and climate change [122] have become more prominent in public health and education policy [4,123,124]. Many policy and program gaps in our food systems have been identified [4,11,123] based on these issues, and food literacy frameworks and school farms may be addressing these new foci. In addition, the social and agroecological movements sparked by *La Via Campesina* in the late 20th century, including ideas of food sovereignty, can be seen in recent food education discourse about the need for social engagement in the food systems as well as more critical approaches to food literacy [10,125]. This contemporary historical context may explain why school farms seek to fill critical gaps in our food system by providing students with both individual and systemic food education.

Our scoping review also found that reported impacts of school farms are similar to those described for school gardens. School gardens and other farm-to-school initiatives have been shown to engage students in nutrition, food security, public health, and ecological sustainability [27,126,127] and improve students’ food literacy and self-development, including confidence, resiliency, and self-sufficiency [18,128]. While the processes of experiential education and interacting with outdoor spaces are similar, and the farms and gardens provide hands-on opportunities to learn about the food system and develop food literacy, this scoping review revealed novel characteristics of school farms that are distinct from other types of outdoor food education programs. School farms can be distinguished from other food literacy interventions by their vocational agriculture training and/or scale of production, which is prevalent across the literature. Specifically, school farms focus on animal husbandry, crop production, and the whole food system rather than small-scale gardening or just food preparation and consumption. This supports the idea that school farms provide a unique educational experience that can go beyond building food literacy. School farms give students opportunities for experiential food education, and prepare students with the professional skills, knowledge, and attributes to be employable within the agricultural sector.

Though research is limited, school farms are postulated in more recent literature to have a range of benefits to students. Some studies suggested that school farms engage diverse populations of students through new food connections and skills to support academic performance [27]. As a means of educating students in a community-based setting, school farms provide an educational experience where students of different backgrounds have the opportunity to learn, share, and excel in skills and knowledge that are otherwise absent in the conventional setting of a formal classroom [10,12,129]. Programs such as school farms call on non-traditional pedagogies, such as land-/place-based and critical food system education, to help fill gaps in traditional food education and develop students’ awareness to build a critical consciousness about one’s role within their food system [12,129,130,131]. Critical and trans-disciplinary learning opportunities, such as those on a school farm, infuse education with a sense of civic obligation and community participation through symbiotic teaching and learning practices shared by diverse agencies, institutions, and people [12]. Participating in transformative environmental and food-based education is critical to promoting a raised consciousness around food system topics and encouraging food and ecological literacy [12,132]. These recent studies on experiential food and environmental learning approaches show how school farms are growing with the food literacy framework in terms of embracing informal food knowledge alongside formal curriculum, and acknowledging and attracting all types of students to engage with their community, place and culture via their food system.

Finally, we noted that the school farm literature to date did not include any references to food literacy interventions involving early care education. To complete the continuum of child education and development, there is growing interest in the US to have farm-to-school programming serve early care and education (ECE) [133,134,135]. Future research is needed to explore agricultural education for children to better define school farms and compare the effects of diverse farming and food education models across school-age groups. We also note that, despite the use of school farm food in school meal programs [23,24,25,26], the literature did not provide consistent or specific structural or quantitative data on the capacity for school farm production and distribution to support school food meal programs. Therefore, it was unclear whether school meal programs were a consistent part of school farms’ programmatic structure. As indicated in the publications in this review [23,24,25,26] and other papers on school meal programs [136,137,138], there is a need for future research on school meal food procurement in relation to school farms. Many other questions remain unanswered, such as what are the effects of school farms on students’ food literacy and their understanding of nutritional health and broader socio-environmental issues. Each community will likely need to define its own goals and roles of a school farm to promote food literacy and support the local food system. Findings of this review indicated some shared characteristics of school farms worldwide; however, school farm programs need to be flexible and diverse to account for local educational, geographic, cultural, and ecological needs.

### Strengths and Weaknesses

This scoping review was limited to three bibliometric databases and did not include a search of school farm organization websites. As a result, more community-based knowledge and program implementation is missing from this review. Although the use of a single search term captured a wide range of literature on food, agricultural, and environmental education that was relevant to school farms, our focus on self-defined school farms meant that we excluded other similar school agriculture and food production education models or school cooperatives [139,140,141] that may have provided further insights into this topic. Also, the lack of synonyms for ‘school farm’ meant that we did not capture post-secondary institutions and the vast majority of school farms in the review address grades K-12 only. Notably, the broad literature on agricultural education programs at post-secondary, university, and college farms that exists [109], however, does not refer to them as school farms. While our search did reveal some publications that included post-secondary farms, this scoping review did not cover the depth of literature that exists for other unique and nuanced programs such as university campus farms. Finally, this scoping review was limited to narrative synthesis and did not support a quantitative analysis of the overall effect of school farms.

A key strength of this review is the systematic process of searching and extracting data to ensure a comprehensive overview of the state of both peer-reviewed and grey literature on self-described school farms published in English. The search included multiple databases of literature covering a wide range of topics, supplemented by hand-searching of reference lists of all included publications. Additionally, multiple independent reviewers separately searched and assessed each record for eligibility, with clear inclusion/exclusion criteria to ensure replicability and rigour.

## 5. Conclusions

School farms have been used worldwide as educational programs to teach students about food systems and engage them in their communities. Most school farms provide essential life skills and behaviours that address health, environmental, and economic issues related to food and agriculture. School farms have the potential to be sustainable food education programs and pedagogical frameworks by expanding core curricula from science and mathematics to include social sciences and humanities courses that students need to graduate from primary and secondary school. Despite growing attention to school farms from over a century of literature, particularly across North America and the UK, empirical evidence is only recent and primarily qualitative. More robust evaluation research is needed that uses mixed methods and community-based approaches to determine the efficacy, experience, and food literacy effects of school farms as experiential food and agricultural education for children and students, and to assess the impact on human and environmental health.

## Figures and Tables

**Figure 1 ijerph-20-05332-f001:**
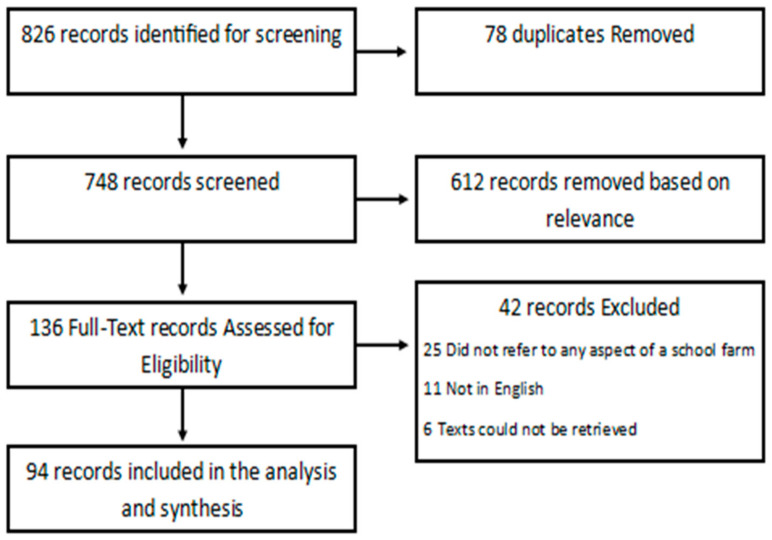
PRISMA flow diagram of search strategy and results.

**Figure 2 ijerph-20-05332-f002:**
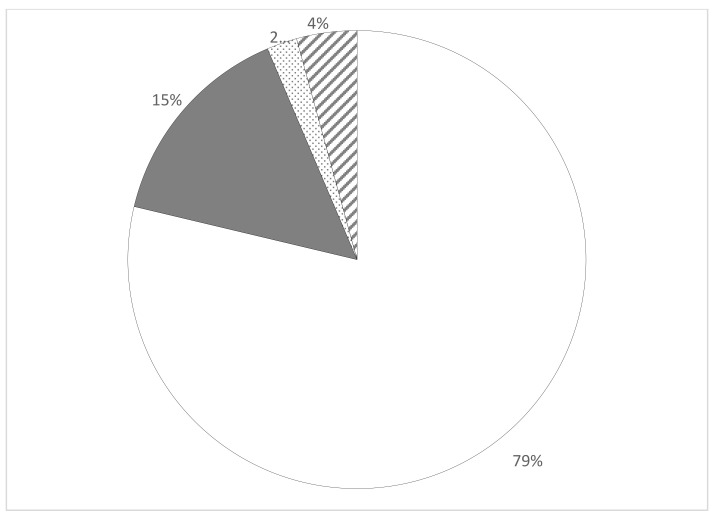
Distribution of school farm literature by type of publication (n = 94). White, editorials and news; Grey, peer-reviewed articles; Dotted, textbooks; Hashed, other (e.g., governmental report, dissertation, etc.).

**Figure 3 ijerph-20-05332-f003:**
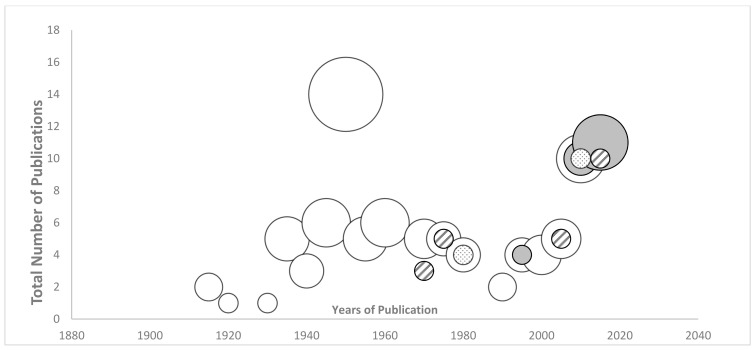
Bubble plot of the frequency and year of publication for each type of publication over a 103-year period. White bubbles, editorials and news; grey bubbles, peer-reviewed original articles; Dotted bubbles, textbooks, Hashed bubbles, other (e.g., governmental report, dissertation, etc.). The *x* axis lists the time period. The *y* axis indicates the total number of publications in that timeframe (n = 94). The size of the bubbles (*z*-axis) indicates the proportion of a given category of publication.

**Table 1 ijerph-20-05332-t001:** Search terms piloting process.

Database	Results
Web of Science	2,046,838
CAB Direct	365,058
Education Source	3,086,829

Search: ((school*) OR (academ*) OR (university*) OR (colleg*) OR (post-secondary) OR (“post-secondary”) OR (campus) AND (farm*)). Searches restricted to English.

**Table 2 ijerph-20-05332-t002:** Summary of the 14 peer-reviewed studies representing the empirical literature on school farms in the scoping review.

Citations	Study Location	Study Design	Objective	Population	Reported Results
Williams and McCarthy 36 (1985) [39]	United States	Case Study	Determine characteristics and benefits of school farms operated by vocational departments in Iowa, Kansas, Missouri, and Nebraska and the characteristics and perceptions of their teachers and school administrators.	68 vocational agriculture departments across Iowa, Kansas, Missouri, and Nebraska	Physical Structure: A majority of school farms were located one mile or less from the classroom, and more than half of them were 10 acres or less in size. Curriculum: To supplement vocational agricultural classrooms and give supervised occupational experience to non-vocational farm students. Obstacle: To make money for the local Future Farmers of America chapter.
Konoshima 43 (1995) [37]	Japan	Case Study	Investigate the agricultural and horticultural activities at school farms supported by the Shiga Prefecture’s local self-governing bodies.	Kindergarten and primary school children	Participants: Most primary schools in Shiga prefecture have participated in agricultural activities at school farms. Obstacle/s: Procuring arable land was the greatest obstacle and most schools borrowed land from neighbouring farms. Curriculum: School farm activities are conducted as domestic science curriculum for first and second graders and then as its own subject for older grades.
Alcock38 (1977) [32]	South Africa	Case Study	Assessing the ‘land school’ and school farm and their effect on the community.	Any farmers that were interested in becoming students	Curriculum: To establish new farmers and improve students’ farming techniques; and to increase student enrolment in farm courses.
Foeken et al. 34 (2010) [23]	Kenya	Case Study	Examine school feeding and school farming in Nakuru Town, Kenya and the extent to which school farming contributes to school feeding programs, and its potential benefits for children.	116 schools (both primary and secondary)	Curriculum: To achieved a degree of self-sufficiency in their School Food programs through school farming. Limited access to land and water, and lack of support and leadership.
Warsh40 (2011) [34]	United States	Historical	Provide a historical report of the Children’s School Farm in New York City from 1902–1931.	School-aged students (not specified)	Curriculum: Focused on the role of nature in urban life and how educators could re-examine the relationship between children, education, and nature.
Wydler35 (2012) [38]	Switzerland	Case Study	Analyse which groups of pupils involved with school farms in Switzerland were interested in different subjects in the eco-educational programming.	28,000 students (1st to 9th grade)	Participants: Majority of students visited the school farm. Curriculum: Farm topics held more appeal for younger students, and girls generally showed more enthusiasm than boys in farm topics. The more times a student visited, the more likely they were to show interest in the farm.
Paffarini et al. 37 (2015) [31]	Italy and Germany	Case study comparison	Identify the main business strategy and principal characteristics of kindergarten farms and examine the value of combining agriculture and education.	2 kindergarten schools (one in Italy and one in Germany)Italy: 14 kids; Germany: 20 kids	Participants: Kindergarten school farm customers are young families unfamiliar with farms. Curriculum: School farms offered educational service and farm products. Obstacles: School farm infrastructure is built on educational and productivity functions carried out by teachers and farm staff. School farm viability is an outcome of revenue streams and costs.
Twenter and Edwards42 (2017) [36]	United States	Historical	Examine the historical evolution of learning spaces and related resources for teaching school-based agricultural education (SBAE) in the United States.	Not specified	Physical structure: Specialized facilities and equipment to educate students in SBAE. Curriculum: Federal legislative mandates solidified occupational training as a part of SBAE. Initially, learning spaces were used for agricultural instruction and production while contemporary spaces integrate academic content with agricultural concepts.
Aniebiat Okon50 (2017) [47]	Nigeria	Comparative	Investigate strategies for school farm land conflict resolution in Akwa Ibom State and understand how to facilitate effective teaching and research in agriculture education to prevent conflict.	150 subjects in study (70 ag teachers, 20 school management staff, 60 community leaders in school communities)	Obstacles: Agricultural education requires land for effective implantation of program. School and community conflict hinder implementation of programs. Solutions: School management and community leaders should communicate to prevent conflict. Government intervention is recommended to resolve school land conflict resolutions.
Corbett et al. 29 (2017) [20]	Australia	Qualitative	Analyse interviews undertaken in 2016 with 22 school farm educators about the state of Tasmania’s school farms.	22 school farm educators from primary and secondary schools)	Curriculum: School farms provide experiential links between food, sustainability, and ecology. They help to address the skills shortage and challenges in retention within the sector. Solutions: School farms respond to workforce needs of the agricultural industry and provide a community-valued environment.
Fifolt et al. 39 (2018) [33]	United States	Case Study	Explore student and parent experiences school-based urban farming with Jones Valley Teaching Farm.	33 students: 29 from grades K-8 and four high school students, and 25 parents	Curriculum: Students on the school farms learned about their own personal and professional interests, developed life skills to make healthier food choices, and agents of change in their community.
Yopp et al. 41 (2018) [35]	United States	Case Study	Observed and interviewed teachers in different in classrooms and laboratories at secondary schools, in livestock barns, greenhouses, and vineyards on school farms and explored the relationship between personal, behavioural, and environmental determinants of social cognitive theory within the total agricultural education program model.	3 secondary school agricultural programs with between 2 and 5 teachers and 1700 and 3100 students.	Curriculum: School farms are made up of three-component agricultural education program model: classroom and laboratory instruction, supervised agricultural experiences (SAE) and a Future Farmers of America (FFA) club. Even in urban settings, school farms were able to use this model and deliver traditional agricultural content. According to teachers was a captivating topic for students and the school farms provided new experience for student who were not familiar with production agriculture.
Lambert et al. 31 (2018) [22]	United States	Descriptive	Explore the characteristics, utilization, perceptions, and potential barriers to using school farms for instructional activities as an experiential learning tool.	64 Oregon agricultural education teachers who identified having a school farm.	Curriculum: School farms provide relevant experiential learning opportunities for students. The primary facilities available on Oregon school farms were for equipment and tool storage and animal projects. Students used school farms for SAE and laboratory instruction. Barriers of successful school farms include the condition of the school farm, facilities, finances, and teachers’ ability to manage and engage all students on the farm.
Fifolt and Morgan32 (2019) [27]	United States	Case Study	Explore principal and teacher experiences with Jones Valley Teaching Farm and how their school farm uses a hands-on food education model to teach academic standards-based lessons.	4600 K-8 students, 20 staff members (15 teachers, 5 principals)	Curriculum: The school farm was seen as a catalyst for student engagement and contributed to the retention of students at risk of dropping out of school. The school farms also created leadership opportunities for students who were not as academically inclined as their peers. The school farm promotes collaboration, communication, and problem-solving skills in their content.

## Data Availability

All relevant data are available in the materials provided.

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
