# Peer review of "What Is a School Farm? Results of a Scoping Review"

_ijerph, 2023, doi:10.3390/ijerph20075332_

Round 1

Reviewer 1 Report

The article is good and clear. It has the merit of having a well-defined goal, which is to review the literature on school farms, analyzing a long period of time. The results are quite surprising, especially it is interesting that there are only 14 peer-reviewed original studies on school farms.

I have some small requests for the authors. In the introduction, I would ask them to add a definition of food literacy, at the beginning, I would say in the third sentence. Immediately after (sorry but there are no numbers near the lines and I can't be more specific), I would explain what "holistic" means.

On the second page there are a couple of repetitions to be eliminated: “experiential learning” in the first two sentences and “diverse students” in the next two sentences. The concepts are repeated identical.

In the Results section, in Figure 2, I'd prefer to use the solid black for the less represented typology, because the big black portion of the pie chart is a bit annoying. But I realize that mine is only an aesthetic, marginal request.

However, Figure 3 is not very understandable. For example, what does the big solid black bubble up in the middle of the graph mean? Why in the same years we have other black bubbles further down? Perhaps an easier to read graph would be preferable.

On page 6 the “National FFA Organization” needs some detail explanation. What is it about?

I would have a similar request for the Zuni people at the top of the next page and for the Young Farmers Club, please add some details such as the nationality of this program.

In Table 2 first row, what do you mean by non-farm students? And on the fifth row, what do you mean by Progressive Era reform? Please, explain. 

In general, for this Table I would have preferred to find more homogeneity in the presentation of the Reported Results, last column. Instead, the subheadings change from one article to another and disappear after a while (p. 11). This makes it difficult to understand the synthesis of the results.

Lines 81-83: it is not clear. If the analysis focuses on school farms for children up to secondary school, how can these schools prepare them for work? Please explain what you mean.

In the Discussion section a difference between school farms for agricultural vocational training and school farms for food literacy and holistic food systems education is perceived, but it is not clearly explained. Please, can you add something to clarify this distinction, if there really is this distinction?

For me that's all.

Reviewer 2 Report

Section “Methods”

I suggest to move the following phrase from methods to results:  Our pilot searches revealed that including all “school” synonyms resulted in unmanageably high numbers of results (Table 1); the majority of records were not relevant to this review

Section “Results”

Figure 2: I suggest to change the pie chart with a histogram because the sum is different from 100%

Table 2: I suggest to standardize the information regarding the population size (i.e number of school, number of children, etc.) and the information as regard the type of curriculum

Section “Discussion”

I suggest to delete this phraseThus, this literature has a strong Anglo-centric/Western bias, and its editorial nature begs the question whether local politics, policies, culture, and educational frameworks also bias the literature”. The research strategy included only English publications.

Section “Strengths and weaknesses”

This is a narrative review with results are not supported by a quantitative analysis. So, I suggest to remark this concept.
